# Diffusion Kurtosis Imaging—A Superior Approach to Assess Tumor–Stroma Ratio in Pancreatic Ductal Adenocarcinoma

**DOI:** 10.3390/cancers12061656

**Published:** 2020-06-22

**Authors:** Philipp Mayer, Yixin Jiang, Tristan A. Kuder, Frank Bergmann, Ekaterina Khristenko, Verena Steinle, Jörg Kaiser, Thilo Hackert, Hans-Ulrich Kauczor, Miriam Klauß, Matthias M. Gaida

**Affiliations:** 1Clinic for Diagnostic and Interventional Radiology, University Hospital Heidelberg, 69120 Heidelberg, Germany; Yixin.Jiang@med.uni-heidelberg.de (Y.J.); Ekaterina.Khristenko@med.uni-heidelberg.de (E.K.); Verena.Steinle@med.uni-heidelberg.de (V.S.); Hans-Ulrich.Kauczor@med.uni-heidelberg.de (H.-U.K.); Miriam.Klauss@med.uni-heidelberg.de (M.K.); 2Medical Physics in Radiology, German Cancer Research Center (DKFZ), 69120 Heidelberg, Germany; T.Kuder@dkfz.de; 3Institute of Pathology, University Hospital Heidelberg, 69120 Heidelberg, Germany; Frank.Bergmann@mail.klinikum-darmstadt.de; 4Clinical Pathology, Klinikum Darmstadt GmbH, 64283 Darmstadt, Germany; 5Department of General, Visceral, and Transplantation Surgery, University Hospital Heidelberg, 69120 Heidelberg, Germany; Joerg.Kaiser@med.uni-heidelberg.de (J.K.); Thilo.Hackert@med.uni-heidelberg.de (T.H.); 6Institute of Pathology, University Medical Center Mainz, JGU-Mainz, 55131 Mainz, Germany; Matthias.Gaida@unimedizin-mainz.de

**Keywords:** pancreatic cancer, diffusion-weighted magnetic resonance imaging, diffusion kurtosis imaging

## Abstract

Extensive desmoplastic stroma is a hallmark of pancreatic ductal adenocarcinoma (PDAC) and contributes to tumor progression and to the relative resistance of tumor cells towards (radio) chemotherapy. Thus, therapies that target the stroma are under intense investigation. To allow the stratification of patients who would profit from such therapies, non-invasive methods assessing the stroma content in relation to tumor mass are required. In the current prospective study, we investigated the usefulness of diffusion-weighted magnetic resonance imaging (DW-MRI), a radiologic method that measures the random motion of water molecules in tissue, in the assessment of PDAC lesions, and more specifically in the desmoplastic tumor stroma. We made use of a sophisticated DW-MRI approach, the so-called diffusion kurtosis imaging (DKI), which possesses potential advantages over conventional and widely used monoexponential diffusion-weighted imaging analysis (cDWI). We found that the diffusion constant D from DKI is highly negatively correlated with the percentage of tumor stroma, the latter determined by histology. D performed significantly better than the widely used apparent diffusion coefficient (ADC) from cDWI in distinguishing stroma-rich (>50% stroma percentage) from stroma-poor tumors (≤50% stroma percentage). Moreover, we could prove the potential of the diffusion constant D as a clinically useful imaging parameter for the differentiation of PDAC-lesions from non-neoplastic pancreatic parenchyma. Therefore, the diffusion constant D from DKI could represent a valuable non-invasive imaging biomarker for assessment of stroma content in PDAC, which is applicable for the clinical diagnostic of PDAC.

## 1. Introduction

Abundant desmoplastic stroma is a hallmark of pancreatic ductal adenocarcinoma (PDAC), accounting for up to 90% of the tumor mass [1].

The desmoplastic stroma is the result of the excessive deposition of extracellular matrix proteins, proliferation of stroma-producing cells, the most prominent being pancreatic stellate cells, and infiltration of immune cells. The stroma alters the microenvironment of the tumor and contributes to tumor progression and to the relative resistance of tumor cells to (radio)chemotherapy as well [2,3,4]. The components of the desmoplastic stroma, such as type I collagen, have been described to enhance proliferation of PDAC cells [5], to increase invasion of tumor cells and to contribute to drug resistance by increasing expression of membrane-type 1 matrix metalloproteinase (MT1-MMP) [6,7]. The tumor-promoting effects of the tumor stroma was not exclusively shown for PDAC, but also for other cancers, such as lung or colon cancer [8,9,10].

Beyond these paracrine effects, the fibrotic stroma may act as physical barrier protecting the PDAC cells from the influx of antitumor drugs [4].

There is an extensive crosstalk between the desmoplastic stroma, immune cells, and the tumor cells [2]. Recent studies indicated that immune infiltrates modulate the stroma composition, which in turn affects progression of PDAC. Both immune infiltrate and stroma content are closely linked to the prognosis of PDAC [4,11,12]. Our previous data indicated that reprogramming of stellate cells by infiltrated polymorphonuclear granulocytes could be a key event in alteration of desmoplastic stroma, which, in turn, may provoke a transition of the tumor towards a more aggressive phenotype [13]. These data were supported by other authors describing and validating a prognostic signature based on leukocyte sub-populations and a stromal composition decisive for the progression free survival [14].

In that, the desmoplastic extracellular matrix (ECM) appears to be a rather promising therapeutic target [15]. Different therapeutic approaches are discussed, which specifically target the tumor stroma rather than the tumor cells itself, and indirectly could contribute to increased sensitivity of tumor cells to chemotherapy [16,17]. New therapy options for PDAC are urgently needed. The primary curative therapy is complete surgical resection, for which less than 20% of patients qualify upfront [18]. Therefore, the vast majority of PDAC patients receive either (neo)adjuvant or palliative cytotoxic chemotherapy, which thus remains the chief treatment option [19]. As with most potent clinical chemotherapy schemes, a combinatorial therapy with 5-fluorouracil (5-FU), leucovorin, irinotecan and oxaliplatin (FOLFIRINOX), FOLFIRINOX significantly increased the survival of PDAC patients compared to the Gemcitabine monotherapy [20]. In patients with advanced stages of the disease, however, modern cytotoxic chemotherapy regimens usually provide only moderate extension of overall survival [19]. Currently, novel personalized treatment strategies using conventional chemotherapeutic approaches in combination with immune checkpoint inhibitors, targets for driver mutations (e.g., kRAS) or interfering with the desmoplastic tumor stroma are under investigation [21]. In 2017, the first US Food and Drug Administration (FDA)-approved treatment targeting non-neoplastic cells, which are present in the tumor stroma, was introduced: The anti-programmed cell death protein 1 (PD-1) monoclonal antibody pembrolizumab was approved as second-line therapy for treatment of a small subgroup of PDAC patients with mismatch repair deficiency [22]. Other therapeutic approaches directly targeting the myo-fibroblastic cells or matrix components, such as the sonic hedgehog pathway, platelet-derived growth factor (PDGF)-receptor, or different proteases to deplete and modulate the architecture of desmoplastic stroma, or the depletion of hyaluronic acid to improve drug delivery revealed beneficial effects [17]. Although the results of a couple of experimental studies on stroma-depleting drugs in PDAC were disappointing, various other therapies targeting or modifying the stroma are currently being investigated [17,22]. However, since the stroma content is highly variable in PDAC, presumably not all patients would benefit from stroma-targeting therapies to the same degree [17]. Especially, the study of Rhim et al. described that a balance between stromal elements and a distinct amount of stroma is necessary to restrain the tumor [23]. Moreover, the aggressiveness of PDAC is caused by the combination of multiple events in the tumor cell itself, in the tumor stroma, and tumor-infiltrating immune cells [24]. An analysis of the desmoplastic stroma relies heavily on histological evaluation on resected surgical specimen. There are, however, numerous studies focused on non-invasive in vivo quantification of fibrosis by different radiological imaging techniques, such as computed tomography (CT) and magnetic resonance imaging (MRI) [25]. CT and MRI are also considered the preferred imaging modalities for the evaluation of a patient with suspected or known PDAC and are used per institutional preference, while other imaging modalities such as endoscopic ultrasound and positron emission tomography (PET)-CT may be used complementarily [26]. CT and MRI have similar sensitivities and specificities for both the diagnosis and vascular involvement of PDAC [27]. Both in contrast-enhanced CT and MRI, the majority of PDACs present as indistinct focal mass with hypoenhancement, most pronounced in the arterial phase [28]. MRI’s unparalleled soft tissue contrast, however, is notably beneficial for detecting small isoattenuating tumors which can be missed by CT [29]. PDACs often show hypointensity in native T1-weighted imaging and variable signal intensity in T2-weighted sequences [28].

Diffusion-weighted magnetic resonance imaging (DW-MRI), a special functional MRI technique, enables the probing of tissue microstructure, thus highlighting the internal characteristics of pancreatic lesions [30] and other tissues. Compared to other radiological imaging techniques, DW-MRI has proven to be particularly useful for quantification of fibrosis in various tissue types [25,31]. DW-MRI measures the random motion of water molecules in the human body. Diffusion is generally more restricted in tissues with high cellular density or dense fibrosis because cell walls, as well as large molecules like collagens, represent physical barriers that impede the free movement of water molecules [32]. In DW-MRI, the tissue is serially imaged with varying degree of water diffusion weighting. The degree of diffusion weighting is referred to as the b-value [33].

In conventional diffusion-weighted imaging (cDWI), diffusion is usually quantitated by the apparent diffusion coefficient (ADC) which is calculated using a monoexponential analysis [32]. The monoexponential model assumes that the diffusion displacement of water molecules follows a Gaussian distribution and, thereby, oversimplifies the diffusion of water molecules in complex biological tissues [33]. Therefore, more sophisticated non-Gaussian diffusion models have been developed, such as the Diffusion Kurtosis Imaging (DKI) that supposedly assesses the complexity of tissue microarchitecture more accurately as compared to cDWI [33]. Indeed, studies by Yoshimaru et al. and Xie et al. showed that DKI might be superior to cDWI for the quantification of liver fibrosis [34,35], and Zhang et al. reported that DKI is a valuable tool for the quantification of pancreatic fibrosis in a chronic pancreatitis rat model [36]. To our knowledge, there are no studies, which evaluate the clinical use of DKI for the analysis of the desmoplastic tumor stroma in human cancers.

Therefore, to provide a non-invasive clinical applicable tool to assess the tumor stroma content in PDAC, this study was conducted.

## 2. Results

### 2.1. Diffusion-Weighted Imaging Analysis

Thirty-one patients with final histopathological diagnosis of PDAC underwent DW-MRI with seven b-values on the day before surgery with curative intent. Moreover, four patients with a final histopathological diagnosis of chronic pancreatitis who underwent the same DW-MRI protocol on the day before partial pancreatic resection were retrospectively analyzed for comparison.

Conventional monoexponential diffusion-weighted imaging analysis (cDWI) was performed, obtaining the apparent diffusion coefficient (ADC) which is a well-established imaging biomarker in oncology [37]. In addition, a more advanced diffusion kurtosis imaging (DKI) analysis was performed in every patient, obtaining the diffusion coefficient D and apparent diffusional kurtosis K. D is analogous to the ADC, but presumed to be the more accurate measure of water diffusion in human tissues, due to its correction for non-Gaussian diffusion behavior [33]. K is a dimensionless statistical parameter which is related to the non-Gaussianity (i.e., the deviation from a normal distribution) of the distribution of water diffusion. K presumably reflects the amount of interfaces within cellular tissues [33]. DW-MRI data were independently evaluated by two readers.

Concerning the group of PDAC patients, agreement between readers was good for ADC values (Intra-class Correlation Coefficient (ICC) range from 0.873 to 0.955), D values (ICC range from 0.868 to 0.890), and K values (ICC range from 0.736 to 0.911). In chronic pancreatitis patients, agreement between readers was moderate for D values (ICC = 0.648) and good for K values (ICC = 0.887) as well as ADC values (ICC = 0.965).

In the following, ADC, D and K values are presented as mean values for both readers if not stated otherwise.

Median ADC, D, and K values with interquartile ranges (IQR) for all PDAC patients are summarized in Table 1. Line diagrams summarizing ADC, D, and K values are presented in Figure 1.

For both diffusion evaluation methods, diffusion was significantly more restricted in tumors than in upstream parenchyma (D_tumor_ versus D_upstream_, *p* < 0.001; ADC_tumor_ versus ADC_upstream_, *p* < 0.001). Differences in water diffusion between tumors and downstream parenchyma were only statistically significant for the diffusion kurtosis analysis (D_tumor_ versus D_downstream_, *p* = 0.008), but not for the conventional monoexponential diffusion analysis (ADC_tumor_ versus ADC_downstream_, *p* = 0.250). K values were not significantly different between tumors and upstream or downstream parenchyma (K_tumor_ versus K_upstream_, *p* = 0.689; K_tumor_ versus K_downstream_, *p* = 0.461).

Receiver operating characteristic (ROC) curves of D, K, and ADC for distinguishing tumors from upstream parenchyma are presented in Figure 2. D showed the highest diagnostic accuracy with an AUC of 0.854 (95% confidence interval (CI): 0.739 to 0.932, *p* < 0.001). The diagnostic accuracy of ADC (AUC 0.765, 95% CI: 0.638 to 0.865, *p* < 0.001) was non-significantly lower than diagnostic accuracy of D (difference between areas 0.089, 95% CI: −0.006 to 0.184, *p* = 0.066). K showed lowest diagnostic accuracy (AUC 0.546, 95% CI: 0.412 to 0.675, *p* = 0.544) which was significantly lower than diagnostic accuracy of D (difference between areas 0.308, 95% CI: 0.122 to 0.494, *p* = 0.001). When the optimal cut-off values of ≤2.282 µm^2^/s for D and ≤1.460 µm^2^/s for ADC were used, sensitivities for distinguishing tumors from upstream parenchyma were 96.8% and 93.6%, and specificities were 69.0% and 55.2%.

Due to the small sample size of patients with downstream parenchyma (*n* = 8), ROC curve analysis was not performed for distinguishing tumors from downstream parenchyma.

Median ADC, D, and K values were non-significantly higher for chronic pancreatitis lesions than for PDAC lesions (1.259 µm^2^/s [IQR 1.202 µm^2^/s to 1.350 µm^2^/s] versus 1.231 µm^2^/s [IQR 1.143 µm^2^/s to 1.340 µm^2^/s), *p* = 0.6408; 1.959 µm^2^/s [IQR 1.899 µm^2^/s to 2.170 µm^2^/s] versus 1.768 µm^2^/s [IQR 1.548 µm^2^/s to 2.073 µm^2^/s], *p* = 0.1949; and 0.907 [IQR 0.681 to 0.967] versus 0.760 [IQR 0.635 to 0.896], *p* = 0.3781). ADC, D, and K values for tumors (D_tumor_, K_tumor_) and chronic pancreatitis lesions (D_cp_, K_cp_) are presented in Appendix A. Example pictures of a chronic pancreatitis patient are shown in Appendix A.

### 2.2. Histopathological Analysis of Tumor Stroma and Tumor Cell Content

The amounts of tumor stromata and tumor cells were evaluated in representative whole tumor tissue sections of each tumor using a software-based approach. The stroma percentage of tumors ranged from 25% to 90% (median 55%, IQR 40% to 80%). Accordingly, the tumor cell percentage ranged from 10% to 75% (median 45%, IQR 20% to 60%).

### 2.3. Correlation of Diffusion-Weighted Imaging Analysis with Histopathological Parameters

There was a significant strong negative rank correlation between D_tumor_ and stroma percentage (r_s_ = −0.852, *p* < 0.001). K_tumor_ and ADC_tumor_ were not significantly correlated to stroma percentage (r_s_ = −0.199, *p* = 0.387, and r_s_ = −0.365, *p* = 0.1034). Tumors with a low stroma percentage ≤50% had significantly higher D_tumor_-values than tumors with high stroma percentage (median 2.047 µm^2^/s, IQR 1.782 µm^2^/s to 2.256 µm^2^/s, versus 1.544 µm^2^/s, IQR 1.383 µm^2^/s to 1.652 µm^2^/s, *p* = 0.001). ADC_tumor_-values and K_tumor_-values did not differ significantly between stroma-poor versus stroma-rich tumors (median ADC_tumor_ 1.260 µm^2^/s, IQR 1.172 µm^2^/s to 1.429 µm^2^/s, versus 1.214 µm^2^/s, IQR 1.124 µm^2^/s to 1.298 µm^2^/s, *p* = 0.260; median K_tumor_ 0.767, IQR 0.605 to 0.899, versus 0.678, IQR 0.617 to 0.812, *p* = 0.526). The dependency of stroma percentage on D is shown in Figure 3.

ROC curves of D_tumor_, K_tumor_, and ADC_tumor_ for predicting high stroma percentage >50% are plotted in Figure 4. The AUC values of D_tumor_, K_tumor_, and ADC_tumor_ were 0.918 (95% CI: 0.714 to 0.992, *p* < 0.001), 0.582 (95% CI: 0.350 to 0.790, *p* = 0.550) and 0.645 (95% CI: 0.410 to 0.839, *p* = 0.254).

When the optimal cut-off value of ≤ 1.664 µm^2^/s for D_tumor_ was used, the sensitivity and specificity for identifying stroma-rich tumors were 81.8% and 90.0%.

Pairwise comparisons of ROC curves showed that D_tumor_ had a significantly better performance in distinguishing stroma-rich from stroma-poor tumors, compared to K_tumor_ (difference between areas 0.336, 95% CI: 0.048 to 0.625, *p* = 0.022) and to ADC_tumor_ (difference between areas 0.273, 95% CI: 0.071 to 0.475, *p* = 0.008).

Example pictures of tumors with high and low stroma percentage are shown in Figure 5 and Figure 6.

ADC_tumor_, D_tumor_, and K_tumor_ values were not significantly different between moderately (G2) and poorly differentiated tumors (G3, *p* ≥ 0.654).

### 2.4. Comparison of Conventional MRI Features and Diffusion Kurtosis Imaging Analysis

The diffusion constant D was at least 0.3 µm^2^/s lower in PDAC lesions than in upstream/downstream parenchyma in 79.3%/75.0% of cases. Visual DWI analysis showed a similar performance with PDAC lesions being hyperintense in DWI with b = 1000 s/mm^2^ compared to upstream/downstream parenchyma in 72.4%/75.0% or cases. Remarkably, some lesions were isointense in DWI with b = 1000 s/mm^2^, but exhibited a diffusion constant D which was distinctly different from the upstream/downstream parenchyma, while in a few other lesions, which were hyperintense in DWI with b = 1000 s/mm^2^, D_tumor_ only slightly differed from D_downstream_/D_downstream_.

T1w hypointensity of PDAC lesions compared to downstream parenchyma (when applicable) was uniformly present. However, only 51.7% of PDAC lesions were T1w hypointense compared to upstream parenchyma. The signal intensity of PDAC lesions in T2-weighted imaging was variable.

Relative signal intensities of PDAC lesions and differences in the diffusion constant D between PDAC lesions and up-/downstream parenchyma are summarized in Appendix A.

## 3. Discussion

In the present study, we investigated whether DW-MRI is useful in clinical routine diagnostics, firstly to predict stroma percentage in PDAC, and secondly to differentiate PDAC from non-neoplastic pancreatic parenchyma, comparing the two methods of conventional monoexponential diffusion imaging analysis (cDWI) and diffusion kurtosis imaging analysis (DKI).

We performed DW-MRI with seven b-values in 31 patients with a final histopathological diagnosis of PDAC on the day before surgery with curative intent. The amount of stroma and tumor cells was evaluated in representative whole tumor sections of the resected tumors using a software-based approach, and the amount of stroma was compared to diffusion parameters from cDWI as well as DKI. This study unravels two important findings, which may have impact to clinical diagnostics.

Firstly, we could establish DKI as a valuable tool for the quantification of stroma content in PDAC and, in this regard, could show that it is superior to the widely clinically used cDWI. We observed a strong significant negative correlation between the diffusion coefficient D_tumor_ from DKI and stroma percentage. Tumors with low stroma percentage ≤50% had significantly higher D_tumor_-values, indicating less restricted diffusion than tumors with high stroma content of >50% of the tumor mass. In contrast, we observed only a weak, non-significant negative correlation between ADC_tumor_ values and stroma percentage. This is in line with findings of Tanaka et al. who analyzed non-neoplastic pancreatic parenchyma and found ADC values to be non-significantly lower in pancreatic specimens with high grade of fibrosis compared to specimens with low grade [38]. Muraoka et al. reported significantly lower ADC values for PDAC specimen with dense fibrosis than with loose fibrosis [39]. However, their loose fibrosis group comprised only three patients [39]. Our results are further supported by a study of Yoshimaru et al. in the context of hepatic fibrosis where DKI analysis was markedly superior to the ADC value for staging of hepatic fibrosis [34].

Hence, the DKI model has the potential to gain further insights into tissue architecture, specifically the tumor–stroma content of PDAC. These data indicate that complicated changes in tissue architecture such as the narrowing of the extracellular spaces and densities by fibrosis are more accurately reflected by D from DKI than by the ADC from cDWI [33]. Similarly, as reported in an animal study assessing hepatic fibrosis by DKI [40], we did not observe significant differences in K values between stroma-rich and stroma-poor tumors. An explanation is that the overall heterogeneity of the tissue microstructure is similar in the stroma compartment and the tumor compartment of PDAC. In contrast, for prostate cancer lesions, an inverse correlation of K values and stromal fraction as well as a positive correlation of D and stromal fraction were reported [41] which can potentially be attributed to the marked structural and architectural differences of prostate and pancreatic cancer lesions, the latter displaying a much higher and extensively inhomogeneous and variable grade of tumor-induced matrix components (e.g., different collagens, proteoglycans, etc.).

The second finding in our patient cohort was that DW-MRI is a useful tool for the differentiation of PDAC from non-neoplastic pancreatic parenchyma. Our findings confirm that diffusion is more restricted in tumors than in non-neoplastic parenchyma, in line with previous reports on DW-MRI in PDAC [39,42,43]. Our apparent diffusion coefficient (ADC) values for PDAC and non-neoplastic pancreatic parenchyma were similar to those reported by Ma et al. [42]. Regarding DKI, our D values from DKI for upstream and downstream parenchyma were similar to those from Kartalis et al. [43]. However, in their study, D values for tumors and K values were somewhat higher than our D_tumor_ and K values [43], which could possibly be attributed to a different choice of b-values. In body imaging, a maximal b-value of 1500 s/mm^2^, as used in our study, is considered appropriate for DKI [33] while Kartalis et al. used a maximal b-value of 1000 s/mm^2^ [43]. This indicates that the results of the DKI analysis of the pancreas can be markedly dependent on the set of b-values used, as reported similarly for prostate imaging [44]. For the optimal choice of b values, both technical and physiological considerations should be taken into account. Obtaining high b-value images with a sufficient signal-to-noise ratio (SNR) is challenging for the evaluation of tissues of the upper abdomen due to the tendency to use short acquisition times to reduce the effect of motion (e.g., respiration) and due to fast signal decays [33]. In contrast, DW-MRI of the prostate might require higher b-values to mitigate the T2 shine-through effect due to its longer T2-relaxation time [41]. Additionally, for some areas of the body, special coil designs, e.g., dedicated breast coils with a high coil count and smaller elements, can increase local sensitivity and decrease noise, thus enabling the acquisition of high b-value images with higher SNR [45]. Moreover, the optimal choice of b values depends on the D value of the studied tissue. Compared to D values of PDAC lesions from the present and a previous study [43], markedly lower D values were reported for breast and prostate cancer lesions which allows for the use of higher b-values in DW-MRI of the breast and prostate with sufficient SNR [46,47]. Standardized pancreatic DW-MRI protocols might help to overcome this limitation in the future.

We cannot absolutely rule out that the high ADC and D values for upstream parenchyma are to a minor extent influenced by partial volume effects from the dilated pancreatic duct although care was taken to exclude the pancreatic duct from the segmentations. However, if this occured, it might have also affected the upstream DWI parameters in previous studies on PDAC [43,48]. The high water diffusivity in the upstream parenchyma in our study may also be in part influenced by pancreatic exocrine products and destruction of pancreatic parenchyma [49].

When comparing cDWI and DKI by applying ROC curve analysis, D from DKI could distinguish tumors from upstream parenchyma with a higher diagnostic accuracy than ADC from cDWI. In line with the study by Kartalis et al., this shows that correction for kurtosis effects could potentially increase the diagnostic accuracy of DW-MRI in PDAC [43]. Indeed, in the present study, not all tumors exhibited D values that were markedly different from non-neoplastic parenchyma and, thus, DKI analysis should be used in combination with analysis of conventional MRI features and visual assessment of DWI. Interestingly, K values were not significantly different between tumors and upstream or downstream parenchyma, implying that microstructural heterogeneity might be similar in these tissues. A limitation of the DKI method is that it is probably of minor use for the differentiation of mass-forming chronic pancreatitis and PDAC due to a significant overlap of D and K values, similarly as reported for the ADC from cDWI in previous studies [50].

In conclusion, our study provides two key findings: Firstly, DKI is a useful tool for the differentiation of PDAC from non-neoplastic pancreatic parenchyma and, secondly, is superior to cDWI for the assessment of tumor–stroma ratio. Thus, DKI could become a valuable imaging biomarker for the non-invasive quantification of the stroma content of PDAC, especially for the preselection of patients for stroma-targeting therapies and therapy surveillance.

## 4. Materials and Methods

The study was approved by the ethics committee of Heidelberg University (S-044/2012). Informed consent was obtained from all patients before enrollment in the study.

Inclusion criteria were (a) a focal solid lesion of the pancreas suspicious for PDAC detected in a previous computed tomography (CT) or magnetic resonance (MR) examination, (b) scheduled surgery for this pancreatic lesion, (c) no neoadjuvant (radio-)chemotherapy. Exclusion criteria were the presence of “MR unsafe” metallic implants and definite histopathological diagnosis other than PDAC.

Between February 2017 and September 2019, 36 patients were enrolled in this prospective study and DW-MRI with 7 b-values (for details see below) was performed in every patient on the day before surgery with curative intent.

Four patients with tumors with final histopathological diagnosis other than PDAC (two adenosquamous carcinomas, one acinar cell carcinoma, and one pancreatic neuroendocrine tumor) were excluded from the analysis. For comparison, we decided to retrospectively analyze the DW-MRI data of one patient from the present study and of three patients from a different study collective with final histopathological diagnosis of chronic pancreatitis (2 women, 2 men, age range 46 to 73 years, median 68.5 years) who underwent the same DW-MRI protocol (see below) on the day before partial pancreatic resection. None of these patients had visible calcifications at preoperative CT imaging.

The final histopathological diagnosis was PDAC in 31 patients (13 women, 18 men, age range 50 to 79 years, median age 67 years, IQR 59 to 74 years). Among these, pylorus-preserving pancreaticoduodenectomy was performed in 12 patients, classic Whipple operation in 1 patient, pancreatic left resection with splenectomy in 6 patients, and total pancreatectomy with splenectomy in 2 patients. Resection was not indicated due to intraoperatively detected hepatic metastases in five patients, peritoneal metastases in three patients, paraaortic lymph node metastases in one patient, and progressed liver cirrhosis in one patient. Among the chronic pancreatitis patients, pylorus-preserving pancreaticoduodenectomy was performed in 3 patients and pancreatic left resection with splenectomy in 1 patient.

Among the 21 resected PDAC-patients, 14 patients had moderately differentiated tumors (G2) and 7 patients had poorly differentiated tumors (G3). T-stage was T2 in 15 patients and T3 in 6 patients. N-stage was N0 in 1 patient, N1 in 8 patients, and N3 in 12 patients. No patient had neoadjuvant (radio-)chemotherapy, which potentially changes the stroma architecture.

### 4.1. Magnetic Resonance Imaging

MR imaging was performed using a 1.5 T scanner (Aera, Siemens Medical Solutions, Erlangen, Germany) with a maximum gradient strength of 45 mT/m, a six-element body-phased array coil, and a 24-channel spine array coil.

The imaging protocol comprised breath-hold T1-weighted in/opposed phase imaging in axial orientation (TR = 155 ms; TE = 2.27 and 4.78 ms), Half-Fourier-Acquired Single-shot Turbo spin Echo (HASTE) T2-weighted imaging in axial orientation (TR = 680 ms; TE = 95 ms), HASTE inversion recovery (IR) T2-weighted imaging (TR = 1000 ms; TE = 80 ms) in coronal orientation. Diffusion-weighted images in axial orientation were acquired using a single shot echo-planar imaging (SE2d-EPI) pulse sequence in expiratory breath-hold, similarly to the previously described [51].

The following imaging parameters were used: acquisition matrix = 100 × 84, pixel spacing = 3.5 mm * 3.5 mm, 14 slices, slice thickness = 5 mm; spacing between slices = 0.25 mm, TR = 2400 ms, TE = 65 ms, spectral fat saturation, bandwidth = 2275 Hz/pixel, k-space based parallel imaging technique (GRAPPA) acceleration factor of two, b-values = 0, 100, 500, 750, 1000, 1250, and 1500 s/mm^2^. To avoid motion artifacts, the acquisition was separated into blocks (b0/b100), (b0/b500), …, (b0/b1500). The blocks b0/b750, b0/b1000, b0/b1250, and b0/b1500 were acquired twice to compensate for a reduced signal-to-noise ratio (SNR) with higher b-values. Each block was acquired in a single breath-hold (TA = 24 s).

The choice of b-values was made by taking the diffusion model, the imaged organ, as well as the correlation with stroma content into account. The measured signal attenuation at low b-values (≤200 s/mm^2^) arises partly from microvascular perfusion [52]. For intermediate b-values, the signal attenuation is sensitive to the spatial scale of the size of a cell, and thus is strongly correlated to the cellularity of the examined tissue [52]. High b-values (>1000 s/mm^2^) increase the sensitivity to smaller spatial scale and make DW-MRI suitable for probing the complex microstructure of the extracellular matrix [52]. DKI requires the use of high b-values to ensure the ability of the sequence to measure non-Gaussian diffusion behavior [33]. To perform DKI, at least three b-values are mandatory. However, the acquisition of a larger number of b-values is generally recommended, with at least two b-values both above and below 1000 s/mm^2^ [33]. DKI was initially applied to the brain where b-values should be as high as 2000 s/mm^2^ to capture the non-Gaussian diffusion behavior [53]. Compared to the brain, however, tissues in the abdomen usually show faster T2 decays and faster signal decays at increasing b-values [33]. Thus, the optimal choice of the highest b-value is lower for body applications where a maximum b-value of 1500 s/mm^2^, as in the present study, is considered to be appropriate [33].

### 4.2. Post Processing of Diffusion-Weighted Magnetic Resonance Imaging Data

Post processing was performed using Medical Imaging Interaction Toolkit (MITK) Diffusion software (v2017.07, DKFZ, Heidelberg, Germany).

Two readers (blind) with 2 and 7 years of experience in abdominal imaging independently drew freehand volumes of interest (VOI) on multiple slices surrounding the pancreatic tumors (VOI_tumor_). The accurate anatomical outlines of the lesions were determined with the help of conventional T1-and T2-weighted MR images as well as contrast enhanced CT scans. Larger necrotic areas, calcifications vessels and ducts were excluded from the VOIs. When possible, upstream and downstream pancreatic parenchyma was segmented by both readers (VOI_upstream_, VOI_downstream_).

The signal intensities within each VOI were averaged before calculating the diffusion fits.

Conventional (=monoexponential) diffusion-weighted imaging (cDWI) analysis was performed according to the following equation:(1)$$S_{b}=S_{0}*e^{\left(-b*ADC\right)}$$
where *S*_0_ constitutes the signal without diffusion weighting, S_b_ constitutes the signal at a specific b-value. The apparent diffusion coefficient (ADC) is a measure of the magnitude of diffusion of water molecules within tissue. ADC values for both readers were calculated for tumors (ADC_tumor_), upstream (ADC_upstream_) and downstream parenchyma (ADC_downstream_).

Diffusion Kurtosis Imaging (DKI) analysis was performed according to the following equation:(2)$$S_{b}=S_{0}*e^{\left(-b*D+\frac{1}{6}*b^{2}*D^{2}*K\right)}$$
where *S_b_* is the signal magnitude with diffusion weighting and *S*_0_ is the signal magnitude with no diffusion weighting. D represents a corrected apparent diffusion coefficient (ADC) accounting for non-Gaussian behavior and K is the apparent diffusion kurtosis. In PDAC patients, D- and K-values for both readers were calculated for tumors (D_tumor_, K_tumor_), upstream (D_upstream_, K_upstream_) and downstream parenchyma (D_downstream_, K_downstream_). In chronic pancreatitis patients, ADC, D, and K values were calculated for chronic pancreatitis lesions (D_cp_, K_cp_).

### 4.3. Qualitative assessment of Magnetic Resonance Images

MR images were qualitatively assessed by two readers in consensus, with a time interval of at least five weeks between DKI analysis and qualitative assessment of MR images. Signal intensities of PDAC lesions in native T1-in-phase, native T2-HASTE, and diffusion-weighted imaging (with b = 1000 s/mm^2^) were compared to upstream and downstream parenchyma (if applicable).

### 4.4. Histopathological Analysis

Tissue samples were obtained from the tissue bank of the National Center for Tumor Diseases (NCT, Heidelberg, Germany) in accordance with the regulations of the tissue bank and the approval of the ethics committee of Heidelberg University (no. 206/2005).

The resected tissue specimens were evaluated by a board certified a histopathologist with more than 10 years of experience in surgical pathology of the pancreas. To overcome the tumor heterogeneity the tumor was (sub)totally embedded in paraffin (10 blocks) and HE (Hematoxylin and eosin) sections were performed. Representative slides of PDAC tissue specimens were scanned at 20-fold magnification using the Aperio CS scanner (Leica Biosystems GmbH, Wetzlar, Germany) and stored in the Aperio svs file format, using JPEG compression in eSlide Manager (Aperio). Further evaluation by a pancreas pathologist was performed using the ImageScope Software v12.3.2.8013 (Aperio). Here, areas of tumor stroma and tumor cells were annotated and the proportion of stroma to tumor was determined for further analysis. Percentages were rounded to the nearest 5%.

### 4.5. Statistical Analysis

Statistical data analysis was performed using MedCalc Version 19.2.1 (MedCalc Software, Ostend, Belgium). *p* < 0.05 were considered to be statistically significant. Data are presented as median with interquartile range (IQR). Results of the diffusion-weighted imaging (DWI) analysis from both readers were averaged to create mean values for ADC, D, and K. Inter-reader reliability for ADC, D, and K values was assessed by using the Intra-class Correlation Coefficient (ICC) with 95% confidence intervals (CI) and applying a 2-way ICC with random raters’ assumption reproducibility. Spearman’s rank correlation coefficients *r_s_* were calculated between stroma percentage on the one side and ADC_tumor_, D_tumor_, as well as K_tumor_ on the other side. ADC_tumor_, D_tumor_, and K_tumor_ values were compared a) between tumors with stroma percentage ≤50% and >50%, and b) between moderately differentiated tumors (G2) and poorly differentiated tumors (G3), using a Mann–Whitney U-test. ADC_tumor_ values were compared to ADC_upstream_ and ADC_downstream_ values, D_tumor_ values to D_upstream_ and D_downstream_ values, as well as K_tumor_ values to K_upstream_ and K_downstream_ values, using Wilcoxon rank sum test. A Mann–Whitney U-test was used to compare ADC, D, and K values between PDAC lesions and chronic pancreatitis lesions.

Receiver operating characteristic (ROC) curves were employed (a) to analyze the diagnostic performance of ADC_tumor_, D_tumor_, and K_tumor_ in distinguishing stroma-rich (stroma percentage > 50%) from stroma-poor tumors (stroma percentage ≤ 50%), (b) to analyze the diagnostic performance of ADC, D, and K values to distinguish tumors from upstream parenchyma. The DeLong method [54] was used for comparison of areas under the curves (AUCs). The AUCs with 95% confidence intervals (CIs) were determined. Sensitivities and specificities of the ROC curves were calculated, and the optimal cut-off values were computed.

## 5. Conclusions

Diffusion kurtosis imaging (DKI) analysis possesses potential advantages over conventional monoexponential diffusion-weighted imaging analysis (cDWI) for the assessment of pancreatic ductal adenocarcinoma (PDAC). We show that the diffusion constant D from DKI is a clinically useful imaging parameter for the differentiation of PDAC lesions from non-neoplastic pancreatic parenchyma. Moreover, our data reveal that D can predict stroma content of PDAC with high diagnostic accuracy and, therefore, could be a useful imaging biomarker for patient stratification and treatment monitoring in stroma-targeting therapies of PDAC.

## Figures and Tables

**Figure 1 cancers-12-01656-f001:**
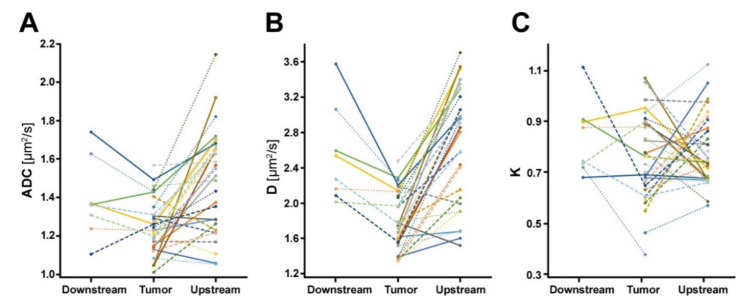
Line diagrams summarizing (**A**) ADC, (**B**) D, and (**C**) K values in PDAC patients. Each line represents one patient. Both ADC and D values were significantly lower in tumors than in upstream parenchyma. Differences in water diffusion between tumors and downstream parenchyma were only statistically significant for D, but not for ADC. K values did not differ significantly between tumors and non-neoplastic parenchyma.

**Figure 2 cancers-12-01656-f002:**
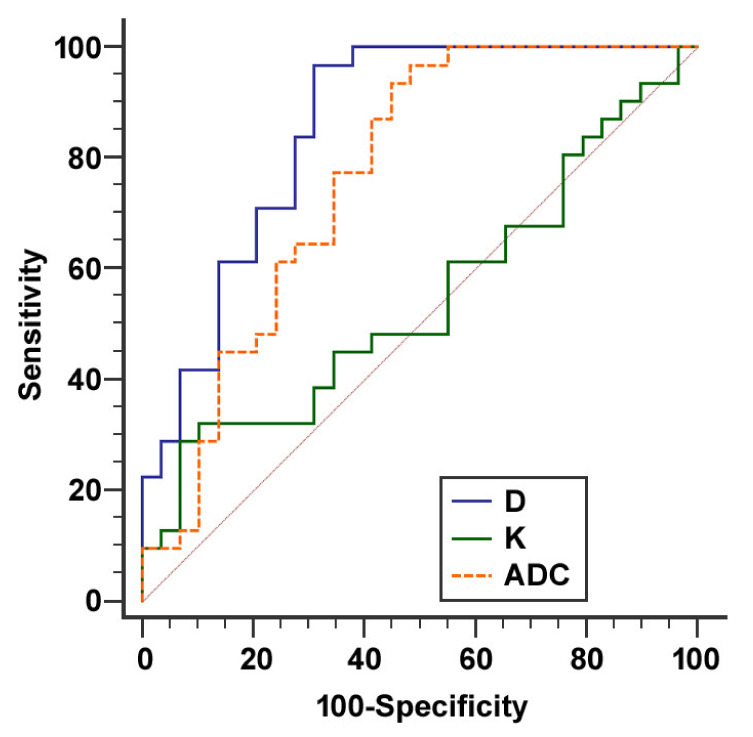
ROC curves for differentiation of tumors from upstream parenchyma using D, K, and ADC. D showed highest diagnostic accuracy with an AUC of 0.854 (95% CI: 0.739 to 0.932, *p* < 0.001).

**Figure 3 cancers-12-01656-f003:**
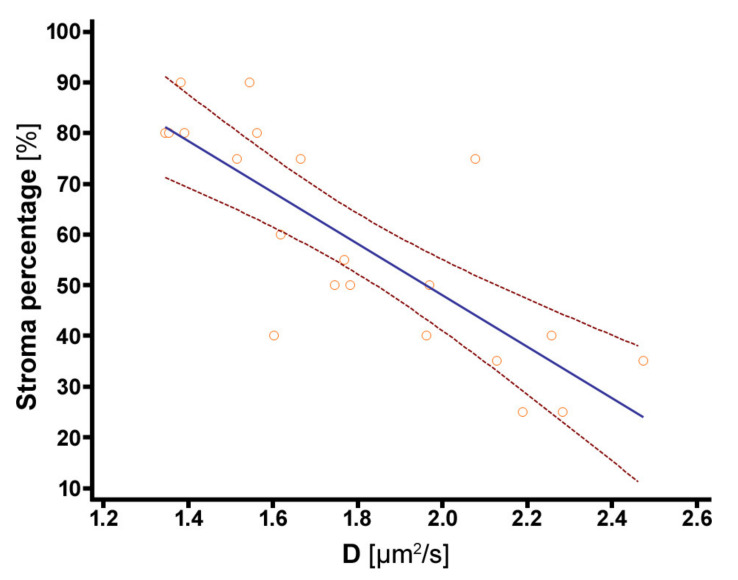
Dependency of stroma percentage on diffusion coefficient D from DKI. Linear regression model: $Stroma\;percentage\left[\%\right]=-50.799\;\left[\frac{\%s}{µm^{2}}\right]*D\left[\frac{µm^{2}}{s}\right]+149.522\;\left[\%\right]$. Goodness of fit: R^2^ = 0.6498. The curves next to the regression line represent the 95% confidence interval. There was a significant strong negative rank correlation between D_tumor_ and stroma percentage (r_s_ = −0.852, *p* < 0.001).

**Figure 4 cancers-12-01656-f004:**
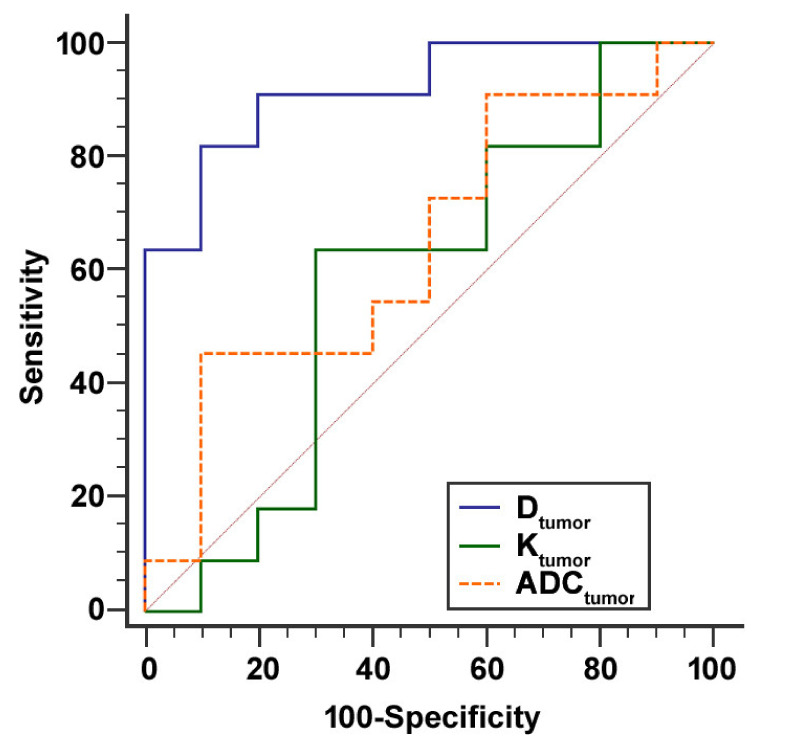
ROC curves for distinguishing stroma-rich from stroma-poor tumors using D_tumor_, K_tumor_, and ADC_tumor_. D_tumor_ had a better performance in distinguishing tumors with stroma percentage >50% from tumors with stroma percentage ≤50%, compared to K_tumor_ and to ADC_tumor_.

**Figure 5 cancers-12-01656-f005:**
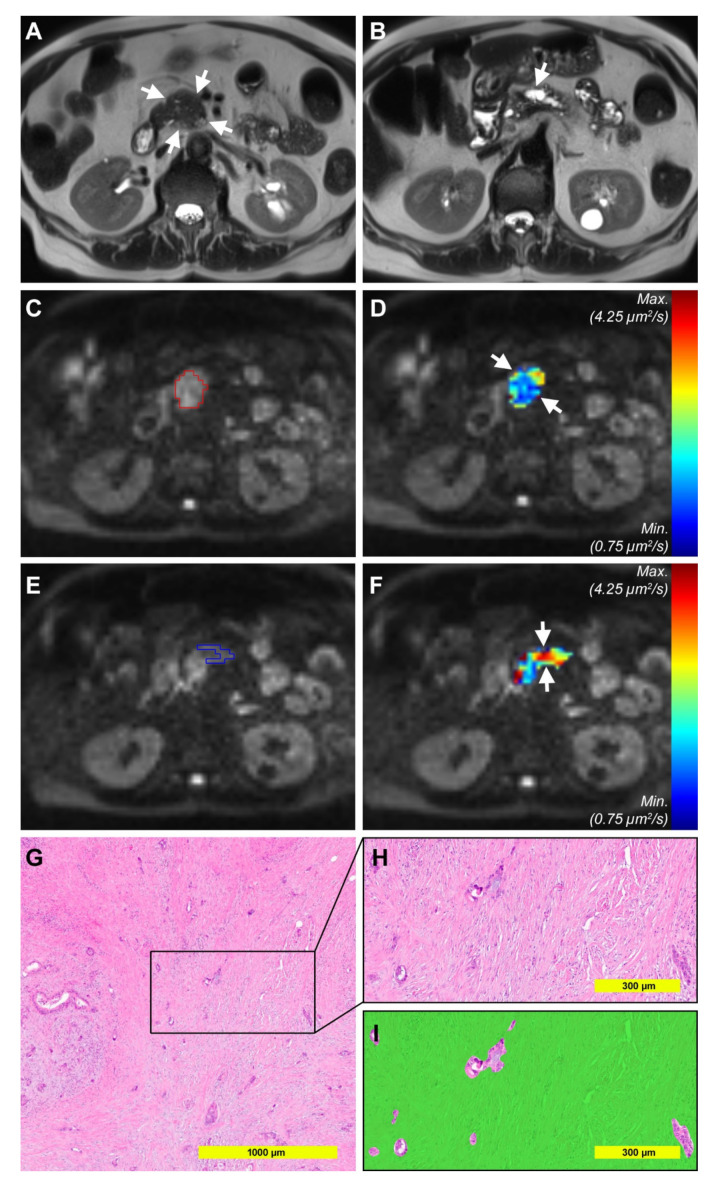
Example pictures of a 74-year-old female patient with high stroma percentage. (**A**) Axial Half-Fourier-Acquired Single-shot Turbo spin Echo (HASTE) T2-weighted image showing the isointense tumor in the pancreatic head (*arrows*). (**B**) Axial T2-weighted image at the level of the pancreatic corpus showing upstream dilatation of the main pancreatic duct (*arrows*). (**C**) Diffusion-weighted image (DWI) with b-value 1000 s/mm^2^ with the volume of interest surrounding the tumor (VOI_tumor_, *red*) from reader 1. (**D**) Corresponding color-coded D-map. Note that the tumor is colored in blue, representing low D values. Mean D_tumor_ for both readers was 1.544 µm^2^/s. (**E**) DWI with b-value 1000 s/mm^2^ with the volume of interest surrounding the upstream parenchyma (VOI_upstream_, *blue*) from reader 1, taking care to avoid the dilated pancreatic duct. Mean D_upstream_ for both readers was 2.855 µm^2^/s. (**F**) Corresponding color-coded D map. (**G**,**H**) Cutouts of a representative HE section from the histopathological specimen. (**I**) Segmented stroma compartment (*green*) shows high stroma percentage (90% for the analyzed area).

**Figure 6 cancers-12-01656-f006:**
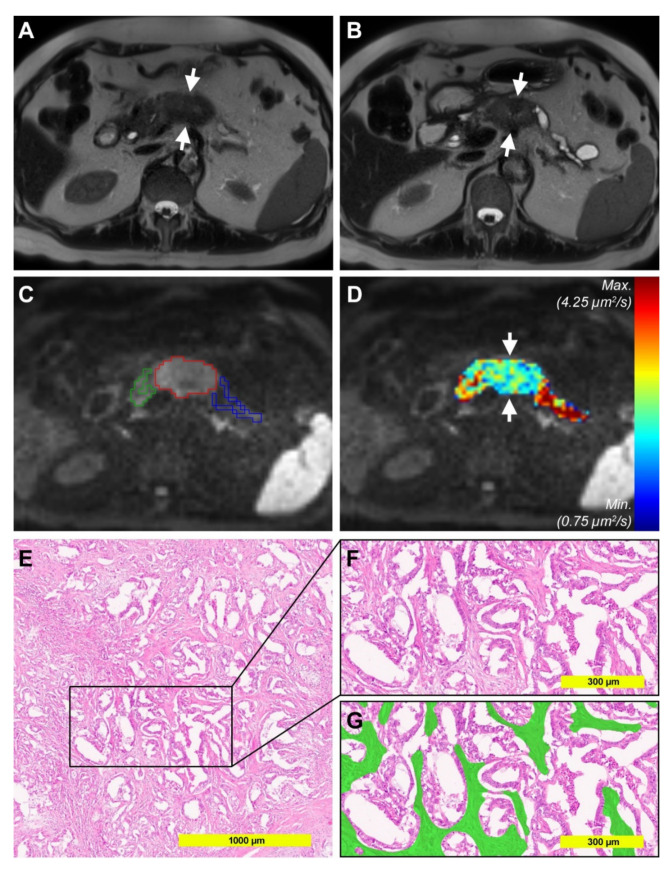
Example pictures of a 75-year-old male patient with low stroma percentage. (**A**) Axial Half-Fourier-Acquired Single-shot Turbo spin Echo (HASTE) T2-weighted image showing the almost isointense tumor in the pancreatic body (*arrows*). (**B**) Axial T2-weighted image at the level of the pancreatic body and tail showing the tumor (*arrows*), upstream dilatation of the main pancreatic duct with concomitant parenchymal atrophy and a cystic lesion in the tail. (**C**) Diffusion-weighted image (DWI) with b-value 1000 s/mm^2^ with volumes of interest surrounding the tumor (VOI_tumor_, *red*), the downstream parenchyma (VOI_downstream_, green), and the upstream parenchyma (VOI_upstream_, blue) from reader 1. (**D**) Corresponding color-coded D map. Note that the tumor is colored in green and blue, representing moderate D values. Mean D_tumor_ for both readers was 2.282 µm^2^/s, mean D_upstream_ was 3.292 µm^2^/s, and D_downstream_ was 2.593 µm^2^/s. (**E**,**F**) Cutouts of a representative HE section from the histopathological specimen. (**G**) Segmented stroma compartment (*green*) shows low stroma percentage (25% for the analyzed area).

**Table 1 cancers-12-01656-t001:** Median ADC, D, and K values with interquartile ranges (IQR) for all PDAC patients.

	Tumor (*n* = 31)	Upstream Parenchyma (*n* = 29)	Downstream Parenchyma (*n* = 8)
**ADC [µm^2^/s]**	1.231 (IQR 1.143 to 1.340)	1.515 (IQR 1.256 to 1.670)	1.362 (IQR 1.273 to 1.499)
**D [µm^2^/s]**	1.768 (IQR 1.548 to 2.073)	2.855 (IQR 2.042 to 3.227)	2.403 (IQR 2.122 to 2.827)
**K**	0.760 (IQR 0.635 to 0.896)	0.746 (IQR 0.673 to 0.881)	0.809 (IQR 0.726 to 0.902)

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
