# Peer review of "Diffusion Kurtosis Imaging—A Superior Approach to Assess Tumor–Stroma Ratio in Pancreatic Ductal Adenocarcinoma"

_cancers, 2020, doi:10.3390/cancers12061656_

Round 1

Reviewer 1 Report

General comments:

In this study, authors explore the use of a novel non-ivasive imaging approach to determine the tumor-stroma ratios in pancreatic ductal carcinoma patients. The work is a prospective study done in a cohort of PDAC patients using post-operative histopatology analyses to validate outcomes. The study concludes that diffusion kurtosis imaging is a useful tool to assess tumor stroma ratios and to differentiate PDAC from normal pancreatric parenchyma.

Although the study present a novel and interesting non-invasive imaging approach, the usefulness of the approach in clinical settings is uncertain/unlikely for the following reasons:

  1. As for today, there are not approved protocols for stratification of pancreatic cancer patients based on tumor-stroma ratios.
  2. As for now, there are not approved therapeutic alternatives targeting the reactive stroma of tumors (only in experimental phases). In fact, therapeutic strategies using stroma-depleting drugs have yielded quite disappointing results hitherto.
  3. Based on the physical principles of the technique, the approach presented by authors will not be able to distinguish between fibrotic pancreatic tissue (like normally observed in chronic pancreatitis and other conditions provoking high-grade fibrotic reactions) from a neoplastic lesion.
  4. Authors claim that DKI is a useful imaging tool for the differentiation of PDAC-lesions from non-neoplastic tissue. However, authors do not compare the approach against other more standard anatomical imaging modalities such as MRI, CT or PET.

Overall, authors present a novel and interesting diagnostic approach. The measurements seem to correlate with more standard tissue-based analytical methods; however, the applicability of the method in the clinics is highly uncertain.

Reviewer 2 Report

In this manuscript, Mayer et al. aimed to investigate the usefulness of diffusion kurtosis imaging as tool to identify PDAC lesions, and to quantify fibrosis in PDAC lesions. Pancreatic ductal adenocarcinoma (PDAC) is estimated to become the second leading cause of cancer-related deaths by 2030, with only 15-20% of patients presenting operable disease at the time of diagnosis. In this regard, the proposed study appears to be of great significance for the scientific medical community.

The manuscript is well written and the experimental data well presented. Nevertheless, have some major comments:

  1. In the “Introduction” section a description about standard and new therapies for treating pancreatic ductal adenocarcinoma (PDAC) is missing. The authors should report these information, and relate them to the aim of their prospective study.

  1. Although desmoplastic stroma is a hallmark of pancreatic ductal adenocarcinoma, and diffusion-weighted magnetic resonance imaging (DW-MRI) has proven to be a useful imaging methodology to quantify fibrosis in the context of desmoplastic stroma, a brief overview about the different available imaging approaches for PDAC imaging should be provided to substantiate the need of the study.

  1. Diffusion kurtosis imaging has been acquired with 7 b values. The authors should provide a brief explanation about why they selected “7 b values” for performing their kurtosis study! How the selection of the b value can influence the results of their imaging analysis?

  1. Is there any dependence between the b value selected and the cellularity/stroma/connective composition of the tumor studied, e.g. PDAC vs PCa or breast cancer? This point need to be explained for those readers are not familiar with DKI-MRI.

  1. For quantitative evaluation, the authors reported that VOIs were drawn surrounding the pancreatic tumors. Did the authors also draw VOIs in tumor-free regions?

Other minor notes for the authors:

Pag2 line 51: “(reviewed in [2–4])”, better to report only the citation numbers.

Pag2 line 51: Please use the correct designation for collagen I “type I collagen”.

Pag 5 Paragraph 2.3: K, D and ADC values are reported without error (or IQR). Please report in the text.
Pag 12 line 343: changing “stroma : tumor” with “stroma to tumor” can make the reading easier.

Round 2

Reviewer 1 Report

Authors have considered seriously the comments raised by reviewers and have introduced amendments that have significantly improved the manuscript.